# Drone Controller Localization Based on RSSI Ratio

**DOI:** 10.3390/s23115163

**Published:** 2023-05-29

**Authors:** Yuhong Wang, Yonghong Zeng, Sumei Sun, Peng Hui Tan, Yugang Ma, Ernest Kurniawan

**Affiliations:** Institute for Infocomm Research, Agency for Science, Technology and Research (A*STAR), Singapore 138632, Singapore; yhzeng@i2r.a-star.edu.sg (Y.Z.);

**Keywords:** localization, RSSI fingerprint, two-ray ground reflection, WLAN channel

## Abstract

We proposed two methods for the localization of drone controllers based on received signal strength indicator (RSSI) ratios: the RSSI ratio fingerprint method and the model-based RSSI ratio algorithm. To evaluate the performance of our proposed algorithms, we conducted both simulations and field trials. The simulation results show that our two proposed RSSI-ratio-based localization methods outperformed the distance mapping algorithm proposed in literature when tested in a WLAN channel. Moreover, increasing the number of sensors improved the localization performance. Averaging a number of RSSI ratio samples also improved the performance in propagation channels that did not exhibit location-dependent fading effects. However, in channels with location-dependent fading effects, averaging a number of RSSI ratio samples did not significantly improve the localization performance. Additionally, reducing the grid size improved the performance in channels with small shadowing factor values, but this only resulted in marginal gains in channels with larger shadowing factors. Our field trial results align with the simulation results in a two-ray ground reflection (TRGR) channel. Our methods provide a robust and effective solution for the localization of drone controllers using RSSI ratios.

## 1. Introduction

Drones have become widely used in many applications such as aerial photography, agriculture, surveillance, etc. However, the increase in the availability and accessibility of high-end drones has created new challenges in security. The unlawful usage of drones has caused threats to key infrastructures such as airports, power plants, airbases, etc. Most of the existing research on counter-drone approaches focuses on the detection and neutralization of intruding drones [1]. However, it is equally important to detect and locate the drone controller in order to counteract the whole intrusion system. 

For drone controller localization, the time difference of arrival (TDoA)-based method was proposed in [2]. In this method, TDoA is measured at different receivers [3,4], which requires all receivers to be synchronous with high accuracy. The authors of [5] localized the drone controller by using the angle of arrival (AoA) combined with triangulation. The authors of [6] proposed the use of a deep learning approach to estimate the AoA and the use of triangulation to locate drone controllers. The methods used in both [5,6] require receivers equipped with antenna arrays to perform the AoA estimation [7]. The TDoA- and AoA-based methods both impose high requirements on hardware. Moreover, in urban and suburban environments with rich multipath propagation without line-of-sight (LOS) paths, the TDoA- or AoA-based methods may not be able to provide accurate results. To address these issues, a received signal strength indicator (RSSI)-based method has been proposed for drone controller localization [8]. In [8], the RSSI of a drone controller signal was measured and transformed into the propagation distance based on a path loss model. Then, a trilateration localization technique was adopted to estimate the location of the drone controller. However, it was found that if the path loss exponent *n* is a piecewise function of distance, ambiguity will occur, and moreover, if the path loss is not a monotonic function of distance, the method presented in [8] cannot be used. The authors of [9] proposed locating an RF source using the RSSI and trilateration. However, their proposed method requires the path loss exponent *n* and the power that will be received at a one-meter distance from the RF source. The authors of [10] proposed locating an RF source using the RSSI and a particle filter. However, this algorithm assumes that the transmitting power of an RF source is constant and known and that the RF source and receiver are always in the LOS of each other. To deal with localization in complicated propagation situations such as indoor environments, the RF-fingerprint-based method has also been widely used in recent decades. The RF fingerprint includes the received signal strength indicator (RSSI) [11,12], channel state information (CSI) [13], or the channel frequency response (CFR) [14]. 

In actual scenarios, the transmission power of drone controllers is unknown, and different drone controllers may transmit at different transmission power levels. To make our localization algorithm insensitive to transmission power, we proposed using the RSSI ratios of different sensors instead of RSSI values to build the fingerprint. We proposed two RSSI-ratio-based localization methods for drone controller localization: the RSSI ratio fingerprint method and the model-based RSSI ratio algorithm. We evaluated the performance of our proposed algorithms via simulations and field trials. Using outdoor channels such as the two-ray ground reflection (TRGR) channel [15] and the WLAN channel [16], we generated an RSSI ratio dataset and presented localization simulation results that demonstrated the impact of the number of sensors, the number of samples, and the grid size on the localization performance. In our field trial, we collected training and test datasets in an outdoor grassland near One North Singapore. Our results regarding the localization performance in the field trial aligned with simulation results obtained in a TRGR channel.

Our main contributions and the importance of these contributions are summarized as follows: First, we proposed two RSSI-ratio-based localization methods that outperformed the distance mapping algorithm proposed in [8] when tested in a WLAN channel. Accurate drone controller localization is crucial to effectively counteract the whole drone intrusion system.Second, the simulation results show that averaging a number of RSSI ratio samples improved the localization performance in propagation channels that did not exhibit location-dependent fading effects. However, in channels with location-dependent fading effects, averaging a number of RSSI ratio samples did not significantly improve the localization performance.Third, the simulation results show that reducing the grid size improved the performance in channels with small shadowing factor values, but this only resulted in marginal gains in channels with larger shadowing factors.Finally, we conducted field trials in outdoor grassland with a Futaba drone controller operating in the frequency-hopping spread spectrum (FHSS) mode and four fixed sensors on the site. Our field trial localization performance aligned with simulation results in a TRGR channel. This indicates that our simulation results are applicable to real-world scenarios and underscores the potential usefulness of the proposed RSSI-ratio-based localization methods for the localization of drone controllers.

Our findings are important for researchers and practitioners who are developing and using localization algorithms in environments with location-dependent fading effects and/or with large shadowing factors.

The advantages of our proposed RSSI ratio fingerprint algorithm are as follows:Our proposed algorithm is based on the RSSI ratio, which does not require complicated hardware. TDoA-based approaches necessitate all sensors to be synchronous with high accuracy, which is prohibitively expensive. AoA-based methods require all sensors to be equipped with an antenna array, which also imposes a high requirement on the hardware. Furthermore, the AoA estimation algorithm is very complex.In urban and suburban environments with rich multipath propagation without dominant LOS paths, traditional TDoA- or AoA-based localization methods may not be able to provide accurate results. However, our proposed algorithm has been shown to achieve accurate results in these types of environments, as verified by our simulation results in WLAN channel F.Our RSSI-ratio-fingerprint-based method does not need to know transmit power or path loss exponent information.

The disadvantage of our proposed RSSI ratio fingerprint algorithm is as follows:During the training stage, data need to be collected to build the training dataset, which is very time-consuming (although in the online localization stage, the complexity is low).

The remainder of this paper is organized as follows: In Section 2, we present a path loss model of a WLAN channel and a TRGR channel. In Section 3, we present our proposed RSSI-ratio-based localization method. In Section 4, we present a simulation evaluation for the WLAN channel and the TRGR channel. In Section 5, we present the results of the field trial. We conclude our work in Section 6.

## 2. Path Loss Model of WLAN and TRGR Channel

In this section, we present the path loss models for the WLAN and TRGR channels. 

### 2.1. WLAN Channel Path Loss Model

WLAN channel F, described in [16], is a multipath fading channel for outdoor environments. By denoting *β* as the path loss, path loss in dB units can be expressed as
(1)$$PL_{WLAN}≜10\log_{10}\left(\beta \right)=\phi_{w}\left(d\right)+z_{w}\left(d\right)$$
where
(2)$$\phi_{w}\left(d\right)=\{\begin{array}{c} 20\log_{10}\left(d\right)+20\log_{10}\left(f_{c}\right)-147.55,\;\;d\leq d_{0}\\ 20\log_{10}\left(d_{0}\right)+20\log_{10}\left(f_{c}\right)-147.55+35\log_{10}\left(\frac{d}{d_{0}}\right),\;\;d>d_{0} \end{array}$$
(3)$$z_{w}\left(d\right)=\{\begin{array}{c} \sigma_{0}X,\;\;d\leq d_{0}\;\\ \sigma X,\;\;d>d_{0}\;\; \end{array}$$
where $f_{c}$ is the carrier frequency, and d is the distance between the transmitter and receiver. $z_{w}\left(d\right)$ denotes the shadowing effect, which is the random variation in signal strength due to the presence of physical obstacles in the transmission path, such as buildings, trees and other structures. X is the Gaussian random variable with zero mean and variance 1. $\sigma_{0}$ and σ are shadowing factors. The shadowing factor is a statistical measure of the amount of variation in signal strength due to shadowing. According to the WLAN channel F model [16],$\;d_{0}$ = 30 m, $\sigma_{0}=3\;dB$ and σ = 6 dB. 

### 2.2. Two-Ray Ground Reflection Channel Path Loss Model

The two-ray ground reflection (TRGR) [15] channel is a propagation model in which the received signal has two components: the LOS component and the reflection component formed by a single ground-reflected wave [15]. Taking the shadowing effect into consideration and denoting β as the path loss, the path loss in dB units for the TRGR channel can be expressed as
(4)$$PL_{TRGR}≜10\log_{10}\left(\beta \right)=\phi_{R}\left(d\right)+z_{R}\left(d\right)$$
where
(5)$$\phi_{R}\left(d\right)=-10\log_{10}\left({\left(\frac{\lambda }{4\pi }\right)}^{2}{\left|\frac{\sqrt{G_{los}}e^{-\frac{j2\pi l}{\lambda }}}{l}+\Gamma \left(\theta \right)\sqrt{G_{gr}}\frac{e^{-\frac{j2\pi \left(l^{\prime }\right)}{\lambda }}}{l^{\prime }}\right|}^{2}\right)$$
(6)$$z_{R}\left(d\right)=\sigma_{R}X_{R}$$
where $l=\sqrt{{\left(h_{t}-h_{r}\right)}^{2}+d^{2}}$ is the length of the LOS ray, $l^{\prime }=\sqrt{{\left(h_{t}+h_{r}\right)}^{2}+d^{2}}$ is the length of the ground reflection ray, and d is the horizontal distance between the transmitter and receiver. $h_{t}$ and $h_{r}$ are the height of the transmitter and receiver, respectively. $G_{los}$ and $G_{gr}$ are the combined antenna gain along the LOS path and reflection path, respectively. λ is the wavelength of transmission, and Γ(θ) is the reflection coefficient with $\theta =\mathrm{actan}\left(\frac{h_{t}+h_{r}}{d}\right)$. In (6), $z_{R}\left(d\right)$ denotes the shadowing effect, $X_{R}$ is the Gaussian variable with zero mean and variance of 1, and $\sigma_{R}$ is the shadowing factor. 

The phase difference between the LOS ray and the reflection ray is $\Delta \phi =\frac{2\pi \left(l^{\prime }-l\right)}{\lambda }$. When distance d is large, then Γ(θ)≈−1 and $G_{los}$ $\approx G_{gr}=G$; in this case, $\phi_{R}\left(d\right)$ can be approximated by
(7)$$\phi_{R}\left(d\right)\approx -10\log_{10}\left({\left(\frac{\lambda \sqrt{G}}{4\pi d}\right)}^{2}{\left|1-e^{j\Delta \phi }\right|}^{2}\right)$$

Clearly, when Δϕ is close to zero, $PL_{TRGR}$ will become a very large value, indicating that a fading effect occurs. The TRGR channel model exhibits fading effects at certain locations where Δϕ approaches zero. In this paper, we call this location-dependent fading effects. In contrast, the WLAN channel F model does not exhibit location-dependent fading effects. Note that in the TRGR channel, the path loss is not a monotonic function of distance d.

## 3. Proposed RSSI-Ratio-Based Localization Method 

In this section, we present our two proposed RSSI-ratio-based localization methods: (1) the RSSI ratio fingerprint localization method and (2) the model-based RSSI ratio algorithm.

### 3.1. RSSI Ratio Fingerprint Method

For the RSSI ratio fingerprint localization method, an RSSI ratio training and test dataset need to be built. Then, a machine learning algorithm is used to estimate the drone controller location.

Assume we need to localize a drone controller in an Ε × Ε area. We divide the whole area into small grids, with each grid being the size of α×α (in our simulation, α is 2.5 m or 1 m), and the whole area is divided into $N_{L}$ = ${\left(|\frac{E}{\alpha }|+1\right)}^{2}$ locations, where ⎣x⎦ denotes the largest integer less than x. Denote $\left(x_{i,\;}y_{i}\right)$ as the coordinates of the *i*-th grid, $i=1,\;2,\;\ldots ,\;N_{L}$. Assume that $N_{r}$ sensors are used for the localization of the drone controller. For each location, we collect $N_{s}$ RSSI samples. Let $r_{i,j}\left(m\right)$ denote m-th ($m=1,\;2,\;\cdots ,\;N_{s}$) RSSI samples received by sensor j at location i ($i=1,\;2,\;\cdots ,\;N_{L})$ in milliwatts. Denote $P_{tx}$ as the drone controller transmission power in milliwatts, which is usually unknown to a drone controller detector. Denote $P_{noise}$ as noise power in milliwatts. $r_{i,j}\left(m\right)$ can be written as
(8)$$r_{i,j}\left(m\right)=P_{rx,i,j}\left(m\right)+P_{noise}$$
where $P_{rx,i,j}\left(m\right)$ is the received signal power in milliwatts. The received signal power is expressed as
(9)$$P_{rx,i,j}\left(m\right)=P_{tx}/\beta_{i,j}\left(m\right)$$
where $\beta_{i,j}\left(m\right)$ is path loss from location i to sensor j. $P_{noise}$ fulfills the following equation:(10)$$10\log_{10}\left(P_{noise}\right)=-174+10\log_{10}\left(BW\right)+NF$$
where BW is the sensor bandwidth, and NF is the sensor noise figure. 

Assume that $P_{rx,i,}\left(m\right)\gg P_{noise}$ so that $P_{noise}$ can be neglected. With this assumption, we obtain
(11)$$10\log_{10}\left(r_{i,j}\left(m\right)\right)\approx \;10\log_{10}(P_{tx})-10\log_{10}\left(\beta_{i,j}\left(m\right)\right)\;=10\log_{10}(P_{tx})-\phi \left(d_{i,j}\right)-z_{m}\left(d_{i,j}\right)$$
where $\beta_{i,j}\left(m\right)$ is the path loss from location i to sensor j, $d_{i,j}=\sqrt{{\left(x_{i}-x_{s,j}\right)}^{2}+{\left(y_{i}-y_{s,j}\right)}^{2}}$ is the distance from location i to sensor j, and ($x_{s,j}$, $y_{s,j})$ is the coordinator of sensor j. $z_{m}\left(d_{i,j}\right)$ denotes the shadowing effect, which is a random variable. $\phi \left(d_{i,j}\right)=\phi_{w}\left(d\right)$ if the propagation channel is the WLAN channel, and $\phi \left(d_{i,j}\right)=\phi_{R}\left(d\right)$ if the propagation channel is the TRGR channel.

Since a drone controller’s transmission power is unknown and is supposed to be time-varying, instead of directly using the RSSI as the fingerprint, we propose using the RSSI ratio as the fingerprint. The m-th RSSI ratio sample between sensor l and sensor k at location i can be written as
(12)$$\begin{array}{l} \gamma_{i,l,k}\left(m\right)=10\log_{10}\left(\frac{r_{i,l}\left(m\right)}{r_{i,k}\left(m\right)}\right)\\ =10\log_{10}\left(r_{i,l}\left(m\right)\right)-10\log_{10}\left(r_{i,k}\left(m\right)\;\right)\\ \approx 10\log_{10}\left(\beta_{i,k}\left(m\right)\right)-10\log_{10}\left(\beta_{i,l}\left(m\right)\right)\\ =\phi \left(d_{i,k}\right)-\phi \left(d_{i,l}\right)+z_{m}\left(d_{i,k}\right)-z_{m}\left(d_{i,l}\right) \end{array}$$
where $\beta_{i,k}\left(m\right)$ and $\beta_{i,l}\left(m\right)$ are the path loss from location i to sensor k and sensor l, respectively, and $d_{i,k}$ and $d_{i,l}$ are the distance from location i to sensor l and sensor k, respectively. 

The RSSI ratio fingerprint at location i is denoted as $\left[(x_{i},\;y_{i}\right),{\;O}_{i}]$, where $\left(x_{i},\;y_{i}\right)$ is the coordinator of location i, and $O_{i}=\left[{\vec{\gamma }}_{i,1,2},{\vec{\gamma }}_{i,1,3},\;\cdots ,{\vec{\gamma }}_{i,1,N},\;{\vec{\gamma }}_{i,2,3},\;{\vec{\gamma }}_{i,2,4},\cdots {\vec{\gamma }}_{i,2,N},\cdots ,\;{\vec{\gamma }}_{i,N-1,N}\right]$. With $N_{r}$ sensors, there are $M=N_{r}\left(N_{r}-1\right)/2$ elements in $O_{i}$, and ${\vec{\gamma }}_{i,l,k}={\left[\gamma_{i,l,k}\left(1\right)\;\gamma_{i,l,k}\left(2\right)\cdots \gamma_{i,l,k}\left(N_{s}\right)\right]}^{T}$. 

With the training dataset and test dataset being collected, a machine learning algorithm such as a DNN (deep neural network), KNN (K-nearest neighbors), or a probability-based method is used to estimate the drone controller location. Let $O=\left[{\vec{\xi }}_{1,2},\;{\vec{\xi }}_{1,3},\;\cdots ,\;{\vec{\xi }}_{1,N},\;{\vec{\xi }}_{2,3},\;{\vec{\xi }}_{2,4},\cdots ,\;{\vec{\xi }}_{2,N},\;\cdots ,\;{\vec{\xi }}_{N-1,N}\right]$ to denote a new observation from the test dataset, where ${\vec{\xi }}_{l,k}$ denotes the RSSI ratio vector with $N_{s}$ samples, and ${\vec{\xi }}_{l,k}={\left[\xi_{l,k}\left(1\right),\;\xi_{l,k}\left(2\right),\;\cdots ,\xi_{l,k}\left(N_{s}\right)\right]}^{T}$. Let $D_{i}$ denote the distance between the new observation O and recorded fingerprint $O_{i}$. $D_{i}$ can be written as
(13)$$D_{i}=\sqrt{\overset{N_{r}-1}{\underset{k=1}{\sum }}\sum_{l=k+1}^{N_{r}}({\bar{\gamma }}_{i,k,l}-{\bar{\xi }}_{k,l}}{)}^{2},\;\;i=1,\;2,\;\cdots ,N_{L}\;$$
where ${\bar{\gamma }}_{i,k,l}=\frac{1}{N_{s}}\overset{N_{s}}{\underset{m=1}{\sum }}\gamma_{i,k,l}\left(m\right)$ is the average of $N_{s}$ RSSI ratio samples for location i, and ${\bar{\xi }}_{k,l}=\frac{1}{N_{s}}\overset{N_{s}}{\underset{m=1}{\sum }}\xi_{l,k}\left(m\right)$ is the average of $N_{s}$ RSSI ratio samples for the new observation:(14)$${\bar{\gamma }}_{i,k,l}=\frac{1}{N_{s}}\sum_{m=1}^{N_{s}}\gamma_{i,k,l}\left(m\right)\;=\phi \left(d_{i,k}\right)-\phi \left(d_{i,l}\right)+\frac{1}{N_{s}}\sum_{m=1}^{N_{s}}z_{m}\left(d_{i,k}\right)-\frac{1}{N_{s}}\sum_{m=1}^{N_{s}}z_{m}\left(d_{i,l}\right)\;=\phi \left(d_{i,k}\right)-\phi \left(d_{i,l}\right)+\nu$$
where $\nu =\frac{1}{N_{s}}\overset{N_{s}}{\underset{m=1}{\sum }}z_{m}\left(d_{i,k}\right)-\frac{1}{N_{s}}\overset{N_{s}}{\underset{m=1}{\sum }}z_{m}\left(d_{i,l}\right)$. ν is a zero mean random variable with variance $\frac{2}{N_{s}}\sigma^{2}$. ${\bar{\xi }}_{k,l}$ can be analyzed in a similar way. Note that, after averaging, the variance in ${\bar{\gamma }}_{i,k,l}$ and ${\bar{\xi }}_{k,l}$ is significantly reduced, which will improve the localization performance.

The nearest neighbor (NN) method selects the location with the minimum Euclidean distance between the current observation and the training dataset [11]:(15)$$\hat{J}=\underset{i}{\mathrm{argmin}}D_{i}$$

The estimation of the drone controller coordinator is ($x_{\hat{J}}$, $y_{\hat{J}}).$ The KNN algorithm selects the K nearest observations and takes the average of the corresponding locations.

### 3.2. Model-Based RSSI Ratio Algorithm

In the RSSI ratio fingerprint localization method, a training dataset needs to be collected, which is time-consuming. In scenarios in which the path loss model is known, we propose using the model-based RSSI ratio algorithm, which does not require a training dataset to be collected. For the model-based RSSI ratio algorithm, we need to know the path loss model of the channel, from which we generate the RSSI ratio. We compare the measured RSSI ratio with a model-generated RSSI ratio and estimate the drone controller’s location based on the minimal distance between the measured RSSI ratio and the model-generated RSSI ratio. Our proposed model-based RSSI ratio algorithm can be described as follows:

Let $O=\left[{\vec{\xi }}_{1,2},\;{\vec{\xi }}_{1,3},\;\cdots ,\;{\vec{\xi }}_{1,N},\;{\vec{\xi }}_{2,3},\;{\vec{\xi }}_{2,4},\cdots ,\;{\vec{\xi }}_{2,N},\;\cdots ,\;{\vec{\xi }}_{N-1,N}\right]$ denote the RSSI ratio vector of a new observation, where ${\vec{\xi }}_{l,k}$ denotes the RSSI ratio vector with $N_{s}$ samples between sensor l and sensor k with ${\vec{\xi }}_{l,k}={\left[\xi_{l,k}\left(1\right),\;\xi_{l,k}\left(2\right),\;\cdots ,\xi_{l,k}\left(N_{s}\right)\right]}^{T}$. Calculate the average of $N_{s}$ RSSI ratio samples using the following equation: ${\bar{\xi }}_{k,l}=\frac{1}{N_{s}}\overset{N_{s}}{\underset{m=1}{\sum }}\xi_{l,k}\left(m\right)$. 

Divide the whole area of interest into $N_{L}$ small grids; the grid size is η×η. Denote $\left(x_{i,\;}y_{i}\right)$ as the coordinates of the i-th grid, $i=1,\;2,\;\ldots ,\;N_{L}$. For wireless channels with location-dependent fading (such as a TRGR channel), we calculate the distance between the measured RSSI ratio and the model-generated RSSI ratio as follows:
(16)$$D_{i}=\sqrt{\overset{N_{r}}{\underset{k=2}{\sum }}({\bar{\gamma }}_{i,k,1}-{\bar{\xi }}_{k,1}}{)}^{2},\;\;i=1,\;2,\;\cdots ,N_{L}$$

For wireless channels without location-dependent fading (such as a WLAN channel), (13) shall be used. Where ${\bar{\gamma }}_{i,k,l}=\phi \left(d_{i,k}\right)-\phi \left(d_{i,l}\right)$ is the RSSI ratio between sensor k and sensor l generated based on the path loss model ($i=1,\ldots ,\;N_{L}$). ϕ(d) is the path loss function, which is a deterministic function of d. $d_{i,k}$ and $d_{i,l}$ are the distance from location i to sensor k and sensor l, respectively. We select the index of the grid that has the smallest RSSI ratio distance as the estimation of location, as illustrated in (15). The estimation of the drone controller coordinator is ($x_{\hat{J}}$, $y_{\hat{J}})$.

The advantage of the model-based RSSI ratio algorithm over the RSSI ratio fingerprint localization method is that the time-consuming training dataset collection can be avoided. However, the path loss function ϕ(d) needs to be known and needs to be accurate enough. In actual implementation, ϕ(d) can be derived using actual field trial measurements. Generally, in order to derive the path loss model from actual trial measurements, the following parameters are required: the transmit power, receiver sensitivity, frequency, distance between transmitter and receiver, antenna heights, obstructions, environmental factors such as weather conditions, temperature, humidity, etc. After the required parameters are collected, either empirical modeling or theoretical modeling can be used to derive the path loss model. The details of these models can be found in [17]. 

## 4. Simulation Evaluation 

In this section, we evaluate the localization performance using the path loss models (1) and (4) for WLAN channel F and the TRGR channel, respectively. The WLAN channel F model is widely used to assess the performance of devices operating in the 2.4 GHz/5 GHz frequency bands in outdoor environments. Most FHSS drone controllers also work in the 2.4 GHz/5 GHz frequency bands; therefore, we selected the WLAN channel F model in our simulation. The TRGR channel model is suitable for modeling flat, open-space areas, which is a pertinent scenario for the operation of drone controllers. The TRGR channel model was adopted in the simulation to compare simulation results with the results of the field trial, which was conducted in a flat, open-space area (Dove Lawn near One North MRT station in Singapore). The WLAN channel F model does not exhibit location-dependent fading effects, whereas the TRGR channel model does. We used both models in our simulation to observe the varying impacts that altering the system parameters had on these channels.

Based on a simulation-generated RSSI ratio dataset, we evaluated the localization performance using the RSSI ratio fingerprint algorithm and the model-based RSSI ratio algorithm. For comparison purposes, the localization performance using the distance mapping algorithm, as described in [8], is also shown for WLAN channel F. The machine learning algorithm for the RSSI ratio fingerprint is KNN (K=10). We investigated the effect of the number of sensors, number of samples, grid size, and shadowing factor on the localization performance. Unless otherwise stated, for the RSSI ratio fingerprint algorithm, the training dataset was generated using a 2.5 m grid size (α=2.5 m), and the drone controller transmitter power $P_{tx}$ was 6.3 milliwatts. For the model-based RSSI ratio algorithm, η equaled 2.5 m. Equations (13) and (16) are used for the WLAN channel and TRGR channel, respectively.

### 4.1. Localization Performance in WLAN Channel F

As shown in Figure 1, we consider a scenario in which sensors are placed on the edges of a 140 m×140 m square (blue square), and the drone controller is randomly positioned inside an 80 m×80 m square (red square). We intentionally designed this layout to ensure that the distance between the sensors and drone controller exceeds 30 m so that the performance of the distance mapping algorithm proposed in [8] can be evaluated without ambiguity (in this condition, the path loss exponent n = 3.5). 

In Figure 2, we present the cumulative distribution function (CDF) of the localization error using different numbers of sensors in WLAN channel F. CDF is defined as $\Phi \left(x\right)=P_{r}\left(X\leq x\right)=\int_{-\infty }^{x}p\left(t\right)dt$, where *X* is the random variable, $P_{r}\left(A\right)$ denotes the probability of an event *A,* and p(x) is the probability density function of *X.* Four sensors (A, B, C, and D), eight sensors (A, B, C, D, E, F, G, and H), and all sixteen sensors are used for the four-sensor, eight-sensor, and sixteen-sensor scenarios, respectively. The number of samples in each location is set to 32. We observe that as the number of sensors increases, the localization performance of all algorithms improves. The RSSI ratio fingerprint algorithm achieves slightly better performance than the model-based RSSI ratio algorithm. Our proposed algorithms significantly outperform the distance mapping algorithm in [8]. With the RSSI ratio fingerprint algorithm, increasing the number of sensors from 4 to 16 reduces the 90th percentile localization error from 10.4 m to 4.32 m, indicating that increasing the number of sensors is an effective way to improve localization performance.

Note that the localization error is not only related to the number of sensors but also related to how the sensors are distributed. With the same number of sensors, the optimal placement of the sensors can reduce the localization error. Interested readers can refer to [18,19] for the optimal placement of sensors.

Figure 3 shows the CDF of the localization error using different numbers of samples for each location in WLAN channel F. In this figure, the number of sensors is four. We can observe that as the number of samples in each location increases, the localization performance of all the algorithms improves. Equation (14) demonstrates that averaging $N_{s}$ samples results in a reduction in the variance in ${\bar{\gamma }}_{i,k,l}$ from $2\sigma^{2}$ to $\frac{2}{N_{s}}\sigma^{2}$, consequently leading to an improvement in the localization performance. Again, our proposed two algorithms outperform the distance mapping algorithm in [8]. With the RSSI ratio fingerprint algorithm, increasing the number of samples from 32 to 128 reduces the 90th percentile localization error from 10.6 m to 5.6 m.

### 4.2. Localization Performance in TRGR Channel 

As shown in Figure 4, we consider a scenario in which sensors are placed on the edges of a 50 m×50 m square, and a drone controller is randomly positioned inside this square. This layout is the same as the layout we used in our field trial test (to be covered in Section 5) so that we could later compare the simulation results with the results from an actual field trial. As in the TRGR channel the path loss is not a monotonic function of distance, the distance mapping algorithm in [8] cannot be used. 

Figure 5 illustrates the CDF of the localization error using different numbers of sensors in the TRGR channel. The number of samples in each location is fixed at 32, and the shadowing factor is set to 0.57. We can observe that as the number of sensors is increased, the localization performance is also improved. In the TRGR channel, the model-based RSSI ratio algorithm achieves slightly better performance than the RSSI ratio fingerprint algorithm. Specifically, regarding the RSSI ratio fingerprint algorithm, increasing the number of sensors from 4 to 16 reduces the 90th percentile localization error from 11.9 m to 2.8 m. On the other hand, regarding the model-based RSSI ratio algorithm, increasing the number of sensors from 4 to 16 reduces the 90th percentile localization error from 10.2 m to 1.8 m. Therefore, this observation indicates that increasing the number of sensors is an efficient way to enhance the localization performance.

Figure 6 shows the CDF of the localization error with different numbers of samples for each location in the TRGR channel. In this figure, the number of sensors is four, and the shadowing factor is 0.57 dB. We can observe that increasing the number of samples from 32 to 128 does not significantly improve the localization performance in the TRGR channel. This is due to the small shadowing factor of 0.57 dB in the TRGR channel, which results in a relatively minor gain from averaging the $N_{s}$ samples. According to (7), the TRGR channel exhibits a location-dependent fading effect, which ultimately limits the localization performance.

Figure 7 presents a comparison of the localization performance with different grid sizes (1 m vs. 2.5 m) under varying shadowing factor levels ($\sigma_{R}$ = 0.57 dB vs. 6 dB) in the TRGR channel using four sensors. The results show that at a shadowing factor level of 0.57 dB, the 90th percentile localization error is 11.9 m and 6.87 m for the 2.5 m and 1 m grid sizes, respectively. Meanwhile, at a shadowing factor level of 6 dB, the 90th percentile localization error is 15.7 m and 14.7 m for the 2.5 m and 1 m grid sizes, respectively. Thus, for small shadowing factor levels (i.e., 0.57 dB), smaller grid sizes provide superior performances. However, for large shadowing factor levels (i.e., 6 dB), smaller grid sizes only provide minimal improvements.

## 5. Drone Controller Localization Field Trial 

This section focuses on the evaluation of the localization performance using data collected during field trials. The field trial area, Dove Lawn, is situated close to the One North MRT station in Singapore. The topography of this area is predominantly flat, and the channel propagation model can be most accurately described using the TRGR model [15].

### 5.1. Data Collection

Figure 8 displays the setup and layout for data collection. Four sensors, namely, W, Z, M, and P, were placed at four corners of a 50 m×50 m square. The sensors utilized to collect data for the drone controller signal were RSA 306B, which is a real-time spectral analyzer manufactured by Tektronix. Each RSA 306B sensor was connected to a laptop using a USB 3.0 interface. The center frequency and bandwidth of each RSA 306B sensor was 2.440 GHz and 40 MHz, respectively. The baseband sampling rate of each RSA 306B sensor was 56 Msps. A central control laptop was used to simultaneously manage the data collection process for all four sensors. All five laptops were connected to a WiFi router, which operated on 5 GHz. The central control laptop ran a data collection server application, while the four other laptops ran a data collection client application. The server application sent a start data collection command, and upon receiving this command, the client application sent a command to the connected RSA 306B sensors to begin data collection and save the collected data to the respective laptop.

The drone controller used in the experiment was the Futaba T14SG, which operates on frequency-hopping spread spectrum (FHSS) technology with a frequency range of 2405.376 MHz to 2477.06 MHz and a hopping frequency interval of 2 MHz. The signal transmitted by the drone controller was a period pulse signal with a 7 ms period and a pulse width of 2 ms.

The collected time domain raw I&Q baseband samples were saved on the laptop for the offline calculation of the RSSI ratio. The RSA 306B sensor was positioned at a height of 1.55 m above the ground. The Futaba T14SG drone controller was positioned 1.2 m above the ground. We gathered data from 231 different locations, with a grid size of 2.5 m, to develop the training dataset. The 231 locations were arranged into eleven rows, with each row containing 21 locations. Furthermore, we acquired raw data from 70 locations to construct the testing dataset. Data were collected on 15 October, 22 October, 29 October, and 5 November 2021 for the training dataset, while the test dataset was collected on 12 November and 19 November 2021.

For location i, denote the I&Q baseband signal of one set collected by sensor k as $x_{i,k}\left(n\right),\;n=0,\;1,\;\cdots ,\;N_{T}-1$, where $N_{T}$ is the number of samples in one set. Divide the I&Q baseband signal into the $L_{c}$ segment, with each segment consisting of $N_{c}=⎣\frac{N_{T}}{L_{c}}⎦$ samples, where ⎣z⎦ denotes taking the largest integer less than z. From the *j*-th segment, the RSSI sample can be calculated as follows:(17)$${\hat{r}}_{i,k}\left(j\right)=\frac{BW_{s}}{BW_{h}}\frac{T_{p}}{T_{w}}\frac{1}{N_{c}}\overset{N_{c}}{\underset{m=1}{\sum }}{\left|x_{i,k}\left(\left(j-1\right)N_{c}+m\right)\right|}^{2},\;j=1,\cdots ,L_{c}$$
where $T_{p}$ and $T_{w}$ are the pulse period and pulse width of the pulse signal transmitted by the drone controller, respectively. For the Futaba T14SG drone controller, the pulse period $T_{p}$ = 7 ms and the pulse width $T_{w}=$ 2 ms. In (17), $BW_{s}$ and $BW_{h}$ are the sensor bandwidth and drone controller hopping bandwidth, respectively. Note that the bandwidth of the RSA 306B sensor is $BW_{s}\;$= 40 MHz and that the hopping frequency bandwidth of T14SG is $BW_{h}=$ 80 MHz. 

As shown in (17), in this paper, RSSI is calculated as the average of the squared magnitude of $N_{c}$ baseband samples in linear scale. Note that, although ${\hat{r}}_{i,k}\left(j\right)$ is proportional to RSSI, they are not technically the same. RSSI calculation mostly depends on the chipset used in commercial communication devices, such as Wi-Fi or Bluetooth. As each vendor has its own method to compute RSSI, this makes RSSI values technology- and/or device-specific. Since we use RSSI ratio for localization, the impact of the difference between ${\hat{r}}_{i,k}\left(j\right)$ and the actual RSSI is reduced.

With $N_{e}$ sets of I&Q baseband signals collected at every location, we are able to obtain $N_{e}L_{c}$ RSSI samples.

### 5.2. Shadowing Factor Analysis Based on Measured Data

The RSSI sample vectors collected by sensor k for location i can be written as
(18)$$\Omega_{i,k}={\left[{\hat{r}}_{i,k}\left(1\right),\cdots ,{\hat{r}}_{i,k}\left(N_{e}L_{c}\right)\right]}^{T}$$

From this equation, we can conclude that
(19)$$\mu_{i,k}=\frac{1}{N_{e}L_{c}}\overset{N_{e}L_{c}}{\underset{j=1}{\sum }}10\log_{10}\left({\hat{r}}_{i,k}\left(j\right)\right)$$
(20)$$\sigma_{i,k}=\sqrt{\frac{1}{N_{e}L_{c}}\overset{N_{e}L_{c}}{\underset{j=1}{\sum }}{\left(10\log_{10}\left({\hat{r}}_{i,k}\left(j\right)\right)-\mu_{i,k}\right)}^{2}}$$
where $\mu_{i,k}$ and $\sigma_{i,k}$ denote the estimation of the mean and standard deviation of all elements of $\Omega_{i,k}$. Note that the units for $\mu_{i,k}$ and $\sigma_{i,k}$ are dBm and dB, respectively. The estimated shadowing factor can be written as
(21)$${\hat{\sigma }}_{shadow}=\frac{1}{N_{L}N_{r}}\overset{N_{r}}{\underset{k=1}{\sum }}\overset{N_{L}}{\underset{i=1}{\sum }}\sigma_{i,k}$$
where $N_{r}$ and $N_{L}$ denote the number of sensors and the number of locations, respectively. The unit of ${\hat{\sigma }}_{shadow}$ is dB.

In our experiment, $N_{r}$=4, and $N_{L}$ = 231. According to the RSA 306B sensor settings, there are $N_{T}$ = 56 e6 I&Q samples per second. We divide one set (which corresponds to a one-second duration) of the I&Q baseband signal into $L_{c}=8$ segments, with each segment consisting of $N_{c}=$7 e6 samples. With $N_{e}=4$ sets of I&Q baseband signal collected by each sensor at every location, each sensor will obtain 32 RSSI samples. Using these 32 RSSI samples, we compute the $\mu_{i,k}$ and $\sigma_{i,k}$ for each location. Figure 9 displays the histogram of $\sigma_{i,k}$ (i=1,⋯,231,k=1,⋯,4). According to (21), our estimated value of shadow factor ${\hat{\sigma }}_{shadow}=$0.5745 dB, which is the mean value of the RSSI standard deviation over 231 locations and 4 sensors.

### 5.3. Comparison between Measured RSSI and Model-Generated RSSI

In this section, we compare the measured RSSI and model-generated RSSI values collected by sensor k for location i. The RSSI sample vectors $\Omega_{i,k}$ collected by sensor k for location i are shown in (18), and the mean value of $\Omega_{i,k}$ is shown in (19). The model-generated RSSI values are derived from the TRGR models (4), (5), and (6), with a shadowing factor, $\sigma_{R}\;$= 0.57 dB. Other parameters used in the path loss calculation for the TRGR model are as follows: $h_{t}=1.2\;m$; $h_{r}=1.55\;m$; λ=0.123 m (carrier frequency is 2440 MHz); vertical polarization; the ground reflection coefficient is calculated as $\Gamma \left(\theta \right)=\frac{\sin\left(\theta \right)-x_{v}}{\sin\left(\theta \right)+x_{v}}$ with $x_{v}=\frac{\sqrt{\varepsilon -\cos{\left(\theta \right)}^{2}}}{\varepsilon }$; and ε=4.5+0.5i is the relative permittivity of dry soil [20]. 

Denote $\Omega_{T,i,k}={\left[r_{T,i,k}\left(1\right),\cdots ,r_{T,i,k}\left(N_{e}L_{c}\right)\right]}^{T}$ as the TRGR-model-generated RSSI samples collected by sensor k for location i; then, the mean value of all elements in $\Omega_{T,i,k}$ can be written as
(22)$$\mu_{T,i,k}=\frac{1}{N_{e}L_{c}}\overset{N_{e}L_{c}}{\underset{j=1}{\sum }}10\log_{10}\left(r_{T,i,k}\left(j\right)\right)$$

Denote $\delta_{i,k}$ (unit is dB) as the difference between $\mu_{T,i,k}$ and $\mu_{i,k}$: (23)$$\delta_{i,k}=\mu_{i,k}-\mu_{T,i,k}$$

Figure 10 illustrates the histogram of $\delta_{i,k}$ (i=1,⋯,231,k=1,⋯,4) between the measured RSSI and the model-generated RSSI. It can be observed that the measured RSSI aligns with the model-generated RSSI. The difference between the measured RSSI and the model-generated RSSI may be attributed to the following: (1) Our use of ε=4.5+0.5i (the relative permittivity of soil) to calculate the reflection coefficient. This value may not be same as the actual relative permittivity of the ground; therefore, the calculated reflection coefficient may not have been the same as the actual reflection coefficient. (2) The antenna radiation pattern that we used in the modeling is a typical dipole antenna radiation pattern, which may not have been same as the actual antenna radiation pattern of the drone controller antenna and the RSA 306B antenna, which were used in our experiment. (3) There were measurement errors when we measured the $h_{t}$, $h_{r}$, and d.

### 5.4. Localization Performance Based on Collected Data

We constructed a training dataset and a testing dataset using the collected data and employed the RSSI ratio fingerprint algorithm to locate the drone controller. The machine learning algorithm of choice was KNN (K = 10). Furthermore, we utilized a model-based RSSI ratio algorithm to determine the drone controller’s location using the TRGR model (4) with a shadowing factor of 0.57 dB.

Figure 11 displays the CDF of the field trial localization error based on the collected data. Two curves are depicted in Figure 11: the red curve corresponds to the RSSI ratio fingerprint algorithm, utilizing both the training and test datasets that are obtained from the collected data. The blue curve corresponds to the model-based RSSI ratio algorithm, only employing the collected test dataset. It can be observed that by using the RSSI ratio fingerprint algorithm, the 90th percentile localization error is 11.32 m, which is consistent with our simulation result (11.9 m). When using the model-based RSSI ratio algorithm, the 70th percentile localization error is 11.1 m. The performance degradation of the model-based RSSI ratio algorithm is due to the mismatch between the actual TRGR path loss model and the path loss model utilized in the simulation, which is caused by the measurement error and discrepancy between the actual antenna radiation pattern and the antenna radiation pattern utilized in the simulation.

### 5.5. Computation, Energy, and Delay Overhead 

During the online localization stage, the complexity of our proposed algorithm is low. In this stage, the sensors begin by collecting data for durations of $N_{e}$ seconds. The time-domain raw I&Q baseband samples are processed subsequently according to (17) and (18) for RSSI samples calculation. To calculate these samples, each connected laptop performs $N_{e}N_{c}L_{c}$ multiplications, $N_{e}\left(N_{c}-1\right)L_{c}$ additions, and $N_{e}L_{c}$ logarithms. Subsequently, the sensors send the RSSI samples to a central control laptop. Upon receiving the RSSI samples from all four sensors, the central control laptop estimates the location according to (13), (14), and (15). For the estimation process, the central control laptop performs $[(N_{s}-1)+N_{r}\left(N_{r}-1\right)/2]N_{L}$ additions and $[N_{r}\left(N_{r}-1\right)/2]N_{L}$ multiplications. Our proposed algorithm is computationally efficient; thus, energy efficiency is high. 

The time from data collection to the acquisition of the localization result is less than $N_{e}+2$ s. 

## 6. Conclusions

We proposed two RSSI-ratio-based methods for the localization of drone controllers: the RSSI ratio fingerprint method and the model-based RSSI ratio algorithm. Compared to the distance mapping algorithm proposed in [8], our methods achieved significantly better performances in WLAN channel F. Our simulation results show that increasing the number of sensors improved the localization performance. Averaging a number of RSSI ratio samples also improved the performance in propagation channels that did not exhibit location-dependent fading effects. However, in channels with location-dependent fading effects, averaging the number of RSSI ratio samples did not significantly improve the localization. Additionally, reducing the grid size improved the performance in channels with small shadowing factor values, but this only resulted in marginal gains in channels with larger shadowing factors. Our field trial location performance results aligned with the simulation results in a TRGR channel. Our methods provide a robust and effective solution for the localization of drone controllers using RSSI ratios. 

## Figures and Tables

**Figure 1 sensors-23-05163-f001:**
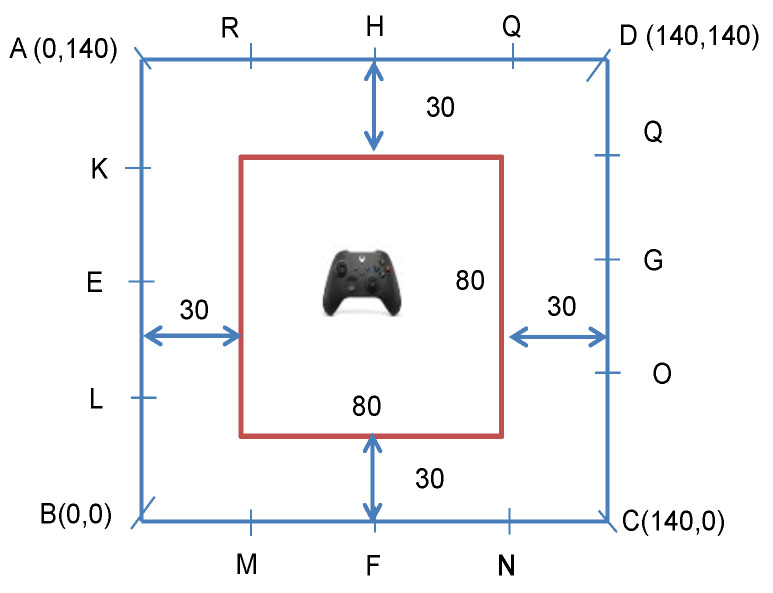
Layout of sensors for localization performance evaluation in WLAN channel F.

**Figure 2 sensors-23-05163-f002:**
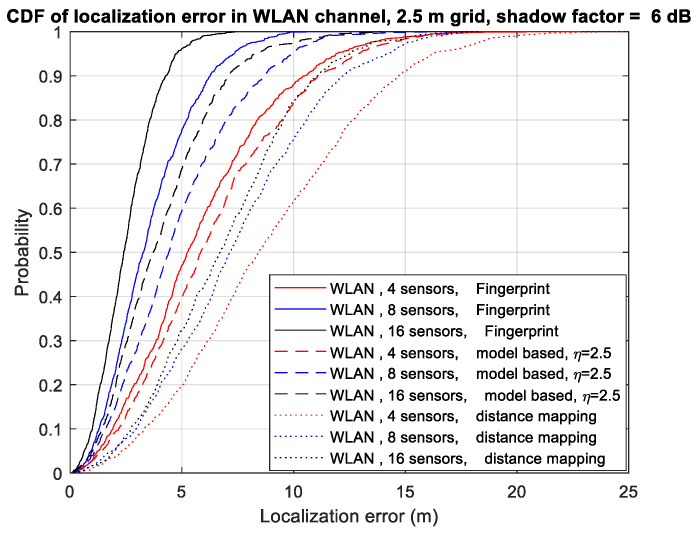
CDF of localization error using different numbers of sensors in WLAN channel F.

**Figure 3 sensors-23-05163-f003:**
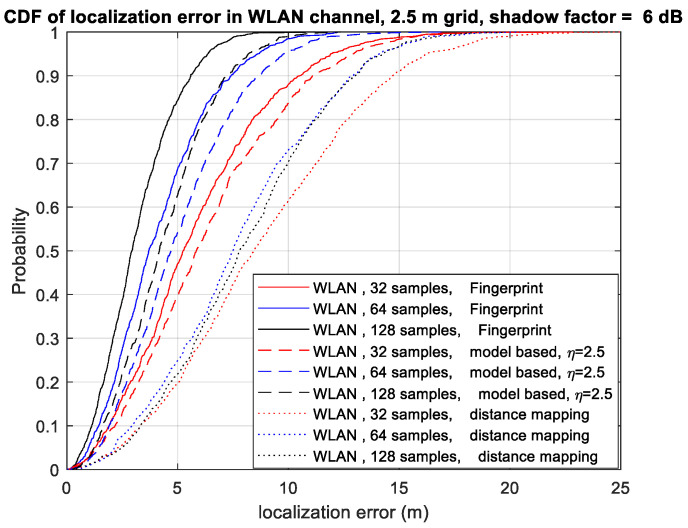
CDF of localization error using different numbers of samples in WLAN channel F.

**Figure 4 sensors-23-05163-f004:**
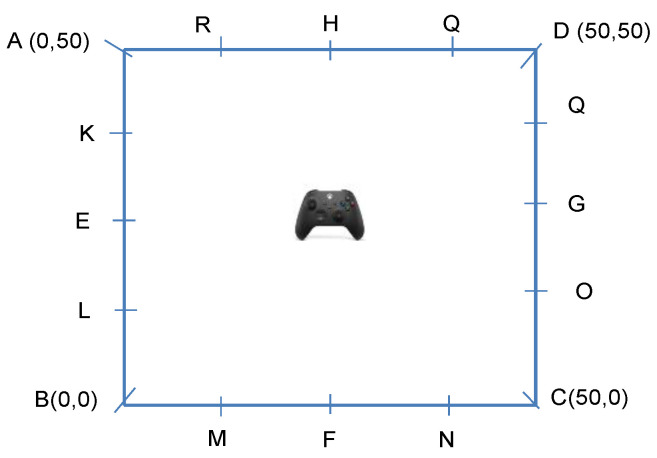
Layout of sensors for evaluation of localization performance in TRGR channel.

**Figure 5 sensors-23-05163-f005:**
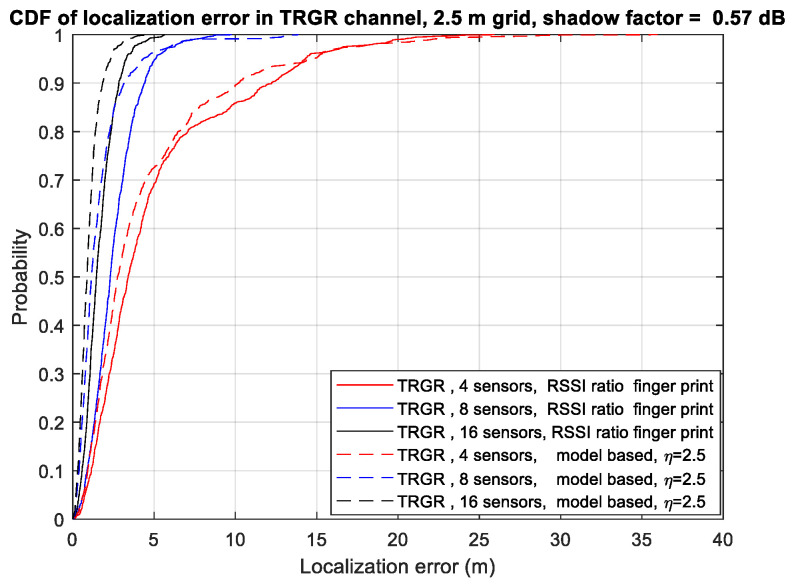
CDF of localization error using different numbers of sensors in TRGR channel.

**Figure 6 sensors-23-05163-f006:**
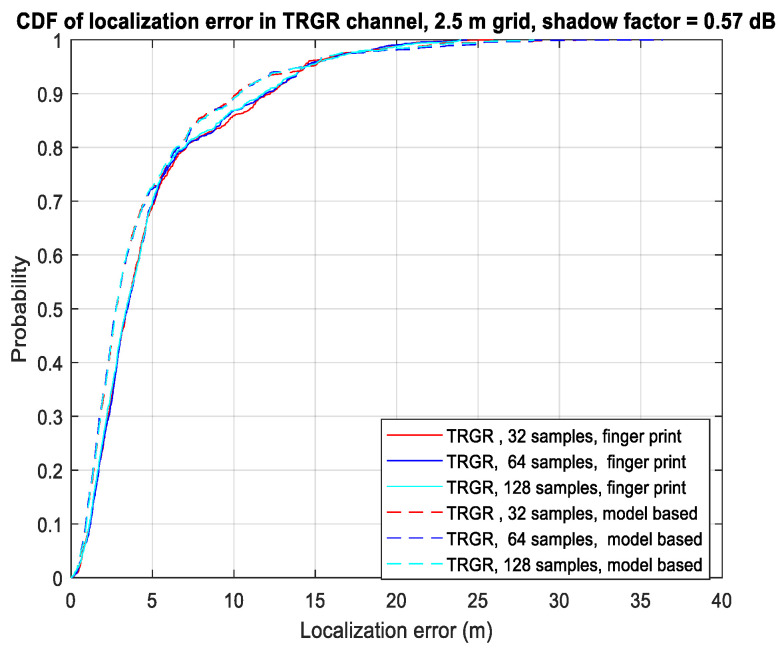
CDF of localization error using different numbers of samples in TRGR channel.

**Figure 7 sensors-23-05163-f007:**
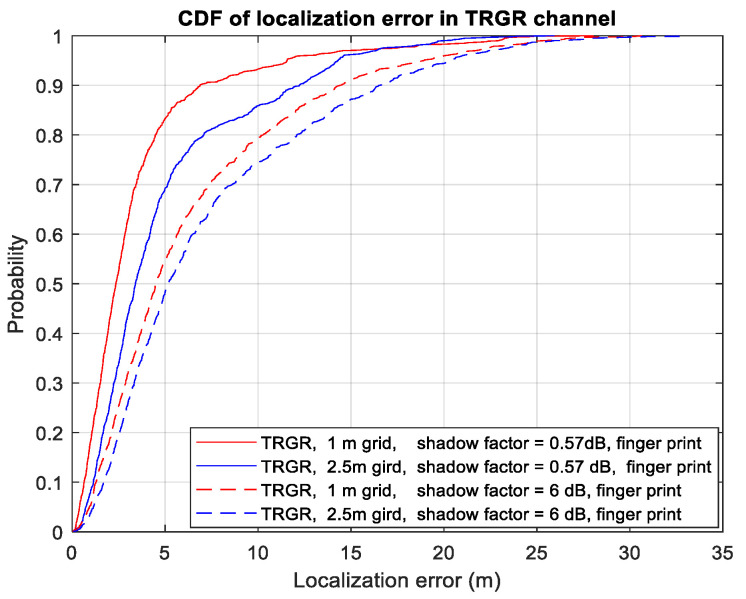
CDF of localization error using different grid sizes in TRGR channel.

**Figure 8 sensors-23-05163-f008:**
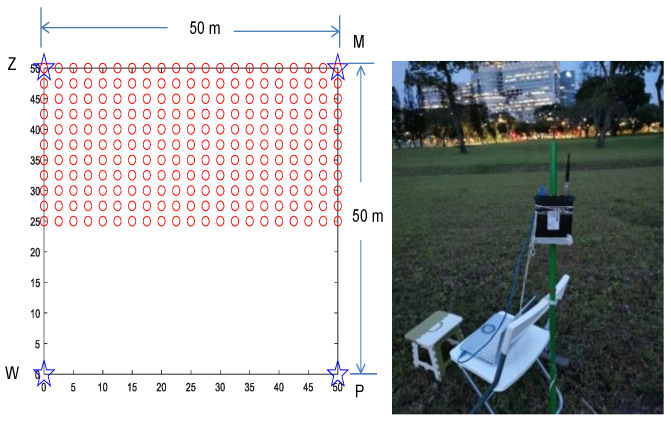
Data collection setup and layout.

**Figure 9 sensors-23-05163-f009:**
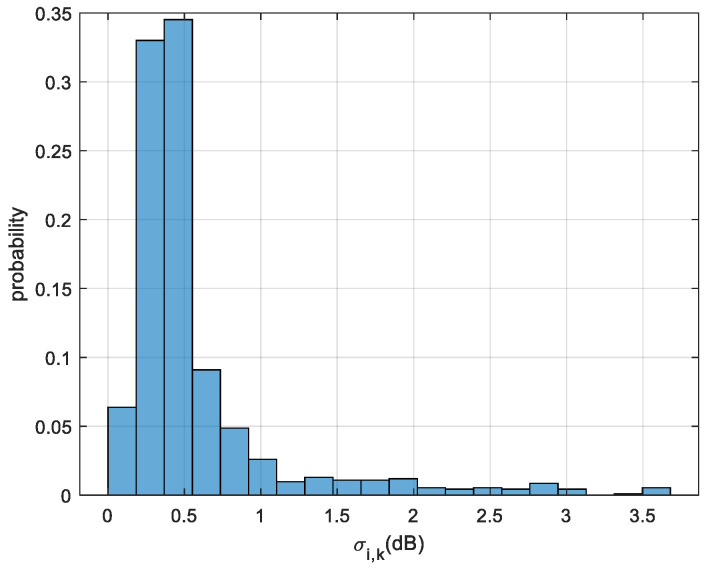
Histogram of $\sigma_{i,k}$ (i=1,⋯,231,k=1,⋯,4) based on measured data.

**Figure 10 sensors-23-05163-f010:**
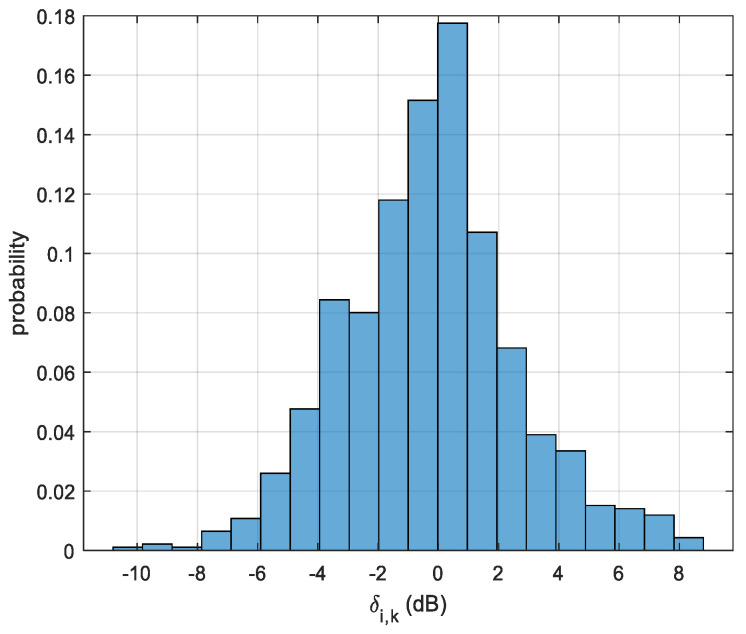
Histogram of $\delta_{i,k}$ (i=1,⋯,231,k=1,⋯,4) between measured RSSI and model-generated RSSI.

**Figure 11 sensors-23-05163-f011:**
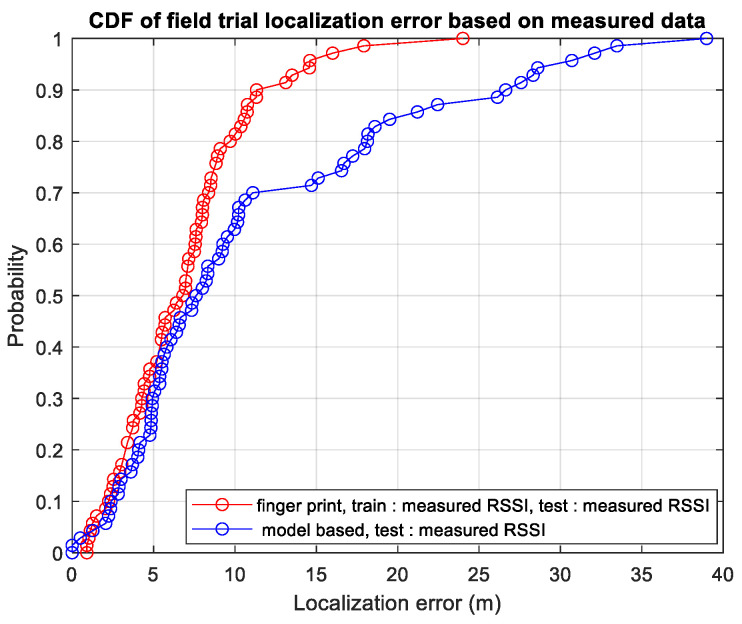
CDF of field trial localization error based on measured data.

## Data Availability

Data available on request due to restrictions. The data presented in this study are available on request from the corresponding author. The data are not publicly available due to ASTAR restrictions.

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
