# Peer review of "Drone Controller Localization Based on RSSI Ratio"

_sensors, 2023, doi:10.3390/s23115163_

Round 1
Reviewer 1 Report
The authors presented a well-written article on the drone localization technique. I suggest to accept the article after some corrections:
1. Please check if the figure's axis CDF and localization error should be exchanged.
2. Please describe the meaning of CDF and shadow sigma terms
3. Figures 3 and 4 are the same.
Reviewer 2 Report
Abstract: Please focus the abstract on your study and your results.
The authors should specify more details regarding the Experiment for the proposed algorithm.
The authors should provide more details regarding the analysis of the results.
I suggest a significant rewrite of the introduction. It should provide an overview of the importance of the main contribution of the proposed algorithm.
The advantage and disadvantages of the work are suggested to be highlighted in comparison with extant studies or methods.
Some syntax errors or improper expressions exist in the manuscript.
More up-to-date studies are suggested to be cited.
Reviewer 3 Report
In this manuscript, the authors propose two UAV controller positioning methods based on the received signal strength index (RSSI) ratio: RSSI ratio fingerprinting and model-based RSSI ratio algorithm. The algorithm proposed in this paper achieves better performance in WLAN channels, and the implementation verifies the conclusions. Therefore, I recommend that after the author corrects the wrong formatting, after which the article can be accepted.
The following are some of my opinions:
1.Drones are mentioned several times in this article, such as Unmanned aerial vehicles (UAVs) and Drone, where it is suggested that the description of drones can be described in one word;
2. Line 144, The writing is a bit cumbersome;
3. To make the article look more canonical, it is recommended that the inserted image and its caption be centered in the text;
4. As shown in Figure 7, it is recommended to enlarge the part of the figure that you want to represent;
5. Line 316, It is recommended to wrap text and pictures in new lines;
6. In the article, Figure 10 and Figure 11 suggest changing to a graphic method to clearly display;
7.At the end of the article, pay attention to the way it is indented, and try to show neatness in the article.
8.The innovation of this paper is only the adoption of RSSl ratio, although the derivation is complete but too simple.
9. This algorithm has too much preparation in the early stage, and the position coordinates2.of each area need to be calculated in advance. This kind of thinking has great constraints on the online use of the positioning system. It is hoped that it can be modified in the initial data preparation.
10. The positioning error of the RSSl ratio method mentioned is relatively large, and it may not be used when the positioning environment fluctuates greatly. ln this regard, is there a better innovative algorithm to reduce the error and improve the applicability.
11. The simulation experiment part only reflects the positioning performance of the proposed algorithm, whether it is possible to show the advantages of the algorithm from more angles.
Reviewer 4 Report
The paper presented two models, a fingerprinting and the TRGR model, using the RSSI ratio method for the drone-controller positioning. Background and motivation are clearly stated in the introduction section. The concept of RSSI ratio is interesting to calibrate the unknown source power. Although numbers of references are cited for various localization techniques, only two references are related to RSSI-based localization, which may be insufficient considering that the RSSI is the manuscript main focus and the RSSI has been widely utilized for decades. Manuscript structure is well-organized. However, the writing structure should be greatly improved due to notations redundancy and parameter inconsistency as well as typos. In this simulation evaluation, the performance was compared in terms of localization error. However, the proposed RSSI ratio itself has not yet been rigorously validation to confirm the correctness of the proposed RSSI ratio model. The source of localization error also has not been discussed in detail, which may not only due to the number of sensors but also how the sensors are distributed. Visualization of the results and measurement/simulation should be greatly improved, especially the unit.
The followings are the specific comments on the paper.
1. The usage and limitation of TRGR and WLAN model F have not been addressed in the manuscript. The authors are expected to provide the applicability of the model to justify why these two models were selected in the simulation.
2. Inconsistency parameter and unit notation:
- Some parameters and units use both italic and straight faces. Please unify the notation
- phi(d) are given two definitions, as in eq. 2 and in eq. 5. Please introduce additional (subscript) notation to differentiate both of them. The same comment also goes for z(d).
- in eq 4, phi(d) and z(d) should not be multiplied, but additive, since both are given in the logarithmic scale.
- parameter tau in line 100 has no definition
- the use of notation i are inconsistency as it is used both superscript and subscript, such as in eq. 9.
- R, P, gamma, xi, and zeta are used as the notation of RSSI or power. The authors are expected to unify the usage of notation for consistency
- eqs. 14a and 14b are also given other equation number as 14 and 15.
3. irrelevant content in Section 2.2: the reviewer feels that the Section 2.2 including figure 1 is irrelevant. Therefore, it can be removed from the manuscript because the path loss characteristic with fading properties of both standard Wireless channel F (stochastic model) and TRGR (deterministic model) are well-known from the literature itself, and thus no need for the discussion.
4. In lines 197-199, the authors discussed the simulation results, which is inappropriate since this section should only address the methodology without any bias from the measurement result.
5. eqs. 14a and 14b should provide the condition of using the respective equations (e.g., for model-based and fingerprinting)
6. In lines 182-193, the reviewer does not see the reason to describe Model-based RSSI ratio in a stepwise manner. For consistency, the authors should use the same writing manner with the explanation of RSSI ratio fingerprint (with the series of equation), or vice versa.
7. In lines 205-206, the reviewer does not fully understand the meaning of derivation of phi(d) from the actual trial measurement. What parameters required to derive from the actual measurement should be clearly stated in this section.
8. In Section 4, validation of the RSSI ratio model in eq. 10 should be rigorously compared in the simulation before evaluation of the localization performance.
9. In Section 5, the authors used the TRGR model for the model-based localization. Although this model in this measurement is valid due to the open-space with less obstacle scenario, the limitation of the TRGR model should be addressed.
10. In Section 5.1, how the RSSI is calculated from the raw IQ data of the 40-MHz signal the via the RSA 306B spectrum analyzer should be clearly explained in this section. Relevant equation or citations may be needed if appropriate.
11. In Section 5.2, the author is wondering whether the method of calculating the shadow sigma is commonly used or not. As the reviewer understand, the shadowing factor is calculated from the distribution of the Rx power measured within a certain area. However, the authors took the mean of standard deviation within the certain area.
Reviewer 5 Report
Title: Drone Controller Localization Based on RSSI Ratio
Strength:
- The authors provided theoretical and experimental evaluation for the performance of the proposed approach.
- The problem description is clear, and the paper is easy to understand.
Weaknesses:
- RSSI-based localization has been extensively studied for different technologies. It is known that RSSI-based localization relatively yields high estimation error due to several factors, including channel dynamics, multipath, and attenuation. The authors presented a promising approach to address the RSSI-based localization drawbacks, but the paper does not compare the proposed approach against other non-RSSI localization techniques.
- Evaluation of the computational, energy, and delay overhead of the proposed approach is not discussed in the paper. Additionally, evaluating the localization performance at different ranges/distances is critical for such an approach.
- Overall, I believe the paper could be significantly improved by conducting experiments to evaluate the performance of the proposed approach in terms of energy, delay, coverage range, and diverse environments (indoor and outdoor).
Round 2
Reviewer 2 Report
Can be accepted
English must be improved
Author Response
Dear Reviewer,
According to reviewer’s comments, we have revised the manuscript (No.2328919, Title: Drone Controller Localization Based on RSSI Ratio).
We would like to thank you for your supporting comments and constructive remarks, which have all been carefully considered to enhance the clarity and quality of this manuscript. We have made every effort to incorporate the recommendations of the Reviewers and improved the structure of this paper. The equation numbers, page numbers and paragraph numbers in this response refer to those in the revised manuscript, unless otherwise stated.
Below is response to reviewers’ comments. The review comments have been given in black-coloured Roman fonts, our responses are in blue-coloured Roman fonts.
In the second round revision, we have enabled the “Track Changes” so that the reviewers and editors can easily follow the changes.
Sincerely,
The Authors
Response to Reviewer 2
Reviewer: Can be accepted, English must be improved.
Authors: Thank you for your time and effort in handling our paper. Many thanks for your valuable comments. During the first round revision, our modified manuscript has been edited by MPDI English Editing Service, thus we believe our English has been improved.
Reviewer 4 Report
Dear Authors,
After carefully checking the authors' report and the revised manuscript, in overall, the reviewer is satisfied with the modification which addressed the major concern to the first submission. However, the reviewer still found a few minors issues that need to be addressed before the reviewer can recommend for the submission.
1. Regarding the responses 14 and 21, the reviewer still feels that the choice of selecting variables are still confusing, which should be improved for better readability.
- the reviewer could not find the reason why the authors define both rij(m) and Rij(m) in the manuscript to express the same quantities but in the different scale (Watt and dBm). Even in eq. 12, the author can directly formulate the equation using only Rij(m).
- Since the subscript tx is used to indicate Ptx as the transmitted power, the reviewer thinks that using Prx is a better choice to represent the received power than introducing another notation as R
- The authors use the subscript dBm to describe the log-scale to Ptx_dBm RdBm. This again is inconsistent because other unit such as Watt and dB has not been used as subscript. Moreover, the use of "_" and "," to separate the subscript dBm is also inconsistent between Ptx_dBm and RdBm. The authors are expected to either address all notation with the "unit subscript", or remove all "unit subscript" from all notation for consistency.
2. definition and notation inconsistency of \beta: The \beta is defined as the path loss in a linear scale (loosely defined as the ratio between the "Tx power (Watt)" and "Rx power (Watt)"). However, the model in eq. 1 implies that the \beta^2 is the path loss, not \beta. Hence, the inconsistency between the notation and definition.
- Again, similar to the 1st comment, the reviewer also thinks that the authors can remove the \beta and uses only \phi (path loss in dB-scale) for all derivation in the manuscript.
3. RSSIi,k (j) in Eq. 19: the reviewer does not understand why the authors decided to define the measured RSSI quantity with RSSIi,k (j) notation whereas other RSSI related quantities summarized in the Response 14 uses a single alphabet letter. The authors are expected to resolve such inconsistency for a better readability to the manuscript.
4. Figure 9, the authors may consider fitting the histogram with log normal to justify the use of shadow factor with the literature.
5. Response 23, The reviewer noticed that the authors intentionally refer RSSI quantity with the received power. Although both quantity are proportional, they are not technically the same. Based on the calculation in eq. 19, the quantity should be referred to as the averaged received power (not the RSSI). Note that the definition of RSSI mostly depends on the chipset used in the commercial communication devices, such as Wi-Fi or Bluetooth. Hence, the authors are expected to explicitly clarify the relationship between RSSI and received power formula used in eq. 19.
Reviewer 5 Report
I am satisfied with this version of the paper. The authors have addressed all my concerns.
Author Response
Dear Reviewer,
According to reviewer’s comments, we have revised the manuscript (No.2328919, Title: Drone Controller Localization Based on RSSI Ratio).
We would like to thank you for your supporting comments and constructive remarks, which have all been carefully considered to enhance the clarity and quality of this manuscript. We have made every effort to incorporate the recommendations of the Reviewers and improved the structure of this paper. The equation numbers, page numbers and paragraph numbers in this response refer to those in the revised manuscript, unless otherwise stated.
Below is response to reviewers’ comments. The review comments have been given in black-coloured Roman fonts, our responses are in blue-coloured Roman fonts.
In the second round revision, we have enabled the “Track Changes” so that the reviewers and editors can easily follow the changes.
Sincerely,
The Authors
Response to Reviewer 5
Reviewer: I am satisfied with this version of the paper. The authors have addressed all my concerns.
Authors: Thank you for your time and effort in handling our paper. Many thanks for your valuable comments.
